# Gradient-free Neural Network Training by Multi-convex Alternating Optimization

## Abstract

In recent years, stochastic gradient descent (SGD) and its variants have been the dominant optimization methods for training deep neural networks. However, SGD suffers from limitations such as the lack of theoretical guarantees, vanishing gradients, excessive sensitivity to input, and difficulties solving highly non-smooth constraints and functions. To overcome these drawbacks, alternating minimization-based methods for deep neural network optimization have attracted fast-increasing attention recently. As an emerging and open domain, however, several new challenges need to be addressed, including 1) Convergence depending on the choice of hyperparameters, and 2) Lack of unified theoretical frameworks with general conditions. We, therefore, propose a novel Deep Learning Alternating Minimization (DLAM) algorithm to deal with these two challenges. Our innovative inequality-constrained formulation infinitely approximates the original problem with non-convex equality constraints, enabling our proof of global convergence of the DLAM algorithm under mild, practical conditions, regardless of the choice of hyperparameters and wide range of various activation functions. Experiments on benchmark datasets demonstrate the effectiveness of DLAM.

## 1 Introduction

Stochastic gradient descent (SGD) and its variants have become popular optimization methods for training deep neural networks. These methods split a dataset into multiple batches and then optimize them sequentially by gradient descent in each epoch. SGD has two main advantages: not only is it simple to implement, but it can also be applied in online settings where new coming training data are used to train models. However, while many researchers have provided solid theoretical guarantees on the convergence of SGD (Kingma & Ba (2014); Reddi et al. (2018); Sutskever et al. (2013)), the assumptions of their proofs cannot be applied to problems involving deep neural networks, which are highly nonsmooth and nonconvex. Aside from the lack of theoretical guarantees, several additional drawbacks restrict the applications of SGD. It suffers from the gradient vanishing problem, meaning that the error signal diminishes as the gradient is backpropagated, which prevents the neural networks from utilizing further training (Taylor et al. (2016)), and the gradient of the activation function is highly sensitive to the input (i.e. poor conditioning), so a small change in the input can lead to a dramatic change in the gradient.

To tackle these intrinsic drawbacks of gradient descent optimization methods, alternating minimization methods have started to attract attention as a potential way to solve deep learning problems. Here, the loss function of a deep neural network is reformulated as a nested function associated with multiple linear and nonlinear transformations across multi-layers. This nested structure is then decomposed into a series of linear and nonlinear equality constraints by introducing auxiliary variables and penalty hyperparameters. The linear and nonlinear equality constraints generate multiple subproblems, which can be minimized alternately. Some recent alternating minimization methods have focused on applying the Alternating Direction Method of Multipliers (ADMM) (Taylor et al. (2016); Wang et al. (2019)) and Block Coordinate Descent (BCD) (Jinshan Zeng (2018)), with empirical evaluations demonstrating good scalability in terms of the number of layers and high accuracy on the test sets, especially for neural networks that are very deep, thanks to parallelism (Taylor et al. (2016); Wang et al. (2019)). For more information, please refer to Section H in the supplementary materials. These methods also avoid gradient vanishing problems and allow for non-differentiable activation functions such as binarized neural networks (Courbariaux et al. (2015)), as well as allowing for complex non-smooth regularization and the constraints that are increasingly important for deep neural architectures that are required to satisfy practical requirements such as interpretability, energy-efficiency, and cost awareness Carreira-Perpinan & Wang (2014).

Table 1: Notations Used in This Paper

| Notations | Descriptions |
|---|---|
| $L$ | Number of layers. |
| $W_l$ | The weight vector in the $l$-th layer. |
| $b_l$ | The intercept in the $l$-th layer. |
| $z_l$ | The temporary variable of the linear mapping in the $l$-th layer. |
| $h_l(z_l)$ | The nonlinear activation function in the $l$-th layer. |
| $a_l$ | The output of the $l$-th layer. |
| $x$ | The input matrix of the neural network. |
| $y$ | The predefined label vector. |
| $R(z_l, y)$ | The risk function in the $l$-th layer. |
| $\Omega_l(W_l)$ | The regularization term in the $l$-th layer. |
| $\varepsilon_l$ | The tolerance of the nonlinear mapping in the $l$-th layer. |

However, as an emerging domain, alternating minimization for deep model optimization suffers from a number of unsolved challenges including: **1. Convergence properties are sensitive to penalty parameters.** One recent work by Wang et al. firstly proved the convergence guarantee of ADMM in the fully-connected neural network problem (Wang et al. (2019)). However, such convergence guarantee is dependent on the choice of penalty hyperparameters: the convergence can not be guaranteed any more when penalty hyperparameters are small. **2. Lack of unified theoretical frameworks with general conditions.** The global convergence of ADMM on deep learning has rarely been explored (Wang et al. (2019); Zeng et al. (2019)). And existing few works are tailored for and limited to few specific loss functions and activation functions: Zeng et al. proved that the ADMM is convergent for square loss function and twice differentiable activation functions (e.g. sigmoid) (Zeng et al. (2019)); Wang et al. proved the convergence of ADMM for the Relu activation function (Wang et al. (2019)). Therefore, there lacks a unified theoretical framework which covers wide range of commonly used losses and activation functions.

In order to simultaneously address these technical problems, we propose a new formulation of the deep neural network problem, along with a novel Deep Learning Alternating Minimization (DLAM) algorithm. The proposed framework is highly generic and sufficiently flexible to be utilized in common fully-connected deep neural network models, as well as being easily extendable to other models such as convolutional neural networks (Krizhevsky et al. (2012)) and recurrent neural networks (Mikolov et al. (2010)). Specifically, we, for the first time, transform the original deep neural network optimization problem into an inequality-constrained problem that can be infinitely approximate to the original one. Applying this innovation to an inequality-constraint based transformation ensures the convexity of all subproblems, and hence easily ensures global minima, the inequality-constraint prevents the output of a nonlinear function from changing much and reduces sensitivity to the input. The operation of matrix inversion is avoided by the quadratic approximation technique and a backtracking algorithm. Moreover, while existing methods require typically strict and complex conditions, such as Kurdyka-ojasiewicz (KL) properties (Lau et al. (2018)) to prove convergence, our proposed method requires simple and mild conditions to guarantee convergence and covers most of the commonly-used loss functions and activation functions, and the choice of hyperparameters has no effect on the convergence of our DLAM algorithm theoretically. Our contributions include:

- We propose a novel formulation for deep neural network optimization. The deeply nested activation functions are disentangled into separate functions innovatively coordinated by inequality constraints that are inherently convex.

- We present a novel and efficient DLAM algorithm. A quadratic approximation technique and a backtracking algorithm are utilized to avoid matrix inversion. Every subproblem has a closed-form solution, further boosting efficiency.

- We investigate several attractive convergence properties of the DLAM algorithm under mild conditions. The model assumptions are very mild, ensuring that most deep learning problems will satisfy our assumptions. The new DLAM algorithm is guaranteed to converge to a critical point.

- We conduct experiments on benchmark datasets to validate our proposed DLAM algorithm. Experiments on two benchmark datasets show that the new algorithm performs well compared with SGD or its variants and ADMM.

The rest of paper is organized as follows. In Section 2, we present the problem formulation and the new DLAM algorithm. In Section 3, we introduce the main convergence results for the DLAM algorithm. Section 4 reports the results of the extensive experiments conducted to validate the convergence and effectiveness of the new DLAM. Section 5 concludes by summarizing the research.

## 2 THE DLAM ALGORITHM

In this section, we present our novel DLAM algorithm. Section 2.1 provides the new algorithm's formulation and Section 2.2 shows how the DLAM algorithm solve all the subproblems.

## 2.1 INEQUALITY APPROXIMATION FOR DEEP LEARNING

The important notations used in this paper are listed in Table 1. A typical fully-connected deep neural network consists of $L$ layers, each of which are defined by a linear mapping and a nonlinear activation function. A linear mapping is composed of a weight vector $W_l \in \mathbb{R}^{n_l \times n_{l-1}}$, where $n_l$ is the number of neurons on the $l$-th layer and an intercept $b_l \in \mathbb{R}^{n_l}$; a nonlinear mapping is defined by an activation function $h_l(\bullet)$. Given an input $a_{l-1} \in \mathbb{R}^{n_{l-1}}$ from the $(l-1)$-th layer, the $l$-th layer outputs $a_l = h_l(W_l a_{l-1} + b_l)$. By introducing an auxiliary variable $z_l$ as the temporary result of the linear mapping, the deep neural network problem is formulated mathematically as follows:

**Problem 1:** $\quad \min_{a_l, W_l, b_l, z_l} R(z_L; y) + \sum_{l=1}^{L} \Omega_l(W_l)$

$$s.t.\ z_l = W_l a_{l-1} + b_l (l=1,\cdots,L),\ a_l = h_l(z_l)(l=1,\cdots,L-1)$$

where $a_0 = x \in \mathbb{R}^d$ is the input of the deep neural network, $d$ is the number of feature dimensions, and $y$ is a predefined label vector. $R(z_L; y) \geq 0$ is the risk function for the $L$-th layer, which is convex and proper, and $\Omega_l(W_l) \geq 0$ is a regularization term on the $l$-th layer, which is also convex and proper. The equality constraint $a_l = h_l(z_l)$ is the most challenging to handle here, because common activation functions such as tanh and smooth sigmoid are nonlinear. This makes them nonconvex constraints and hence it is difficult to obtain a global minimum when updating $z_l$ (Taylor et al. (2016)). To deal with this challenge, we innovatively transform the original nonconvex constraints into convex inequality constraints, which can be infinitely approximate to Problem 1. To do this, we introduce a tolerance $\varepsilon_l > 0$ and reformulate Problem 1 to reach the following form:

$$\min_{W_l, b_l, z_l, a_l} R(z_L; y) + \sum_{l=1}^{L} \Omega_l(W_l) + \sum_{l=1}^{L-1} \mathbb{I}(h_l(z_l) - \varepsilon_l \leq a_l \leq h_l(z_l) + \varepsilon_l)$$

$$s.t.\ z_l = W_l a_{l-1} + b_l (l=1,\cdots,L)$$

$\mathbb{I}(h_l(z_l) - \varepsilon_l \leq a_l \leq h_l(z_l) + \varepsilon_l)$ is an indicator function such that the value is 0 if $h_l(z_l) - \varepsilon_l \leq a_l \leq h_l(z_l) + \varepsilon_l$ and $\infty$ otherwise. For the linear constraint $z_l = W_l a_{l-1} + b_l$, this can be transformed into a penalty term in the objective function to minimize the difference between $z_l$ and $W_l a_{l-1} + b_l$. The formulation is shown as follows:

**Problem 2:** $\quad \min_{W_l, b_l, z_l, a_l} F(\mathbf{W}, \mathbf{b}, \mathbf{z}, \mathbf{a}) = R(z_L; y) + \sum_{l=1}^{L} \Omega_l(W_l)$

$$+ \sum_{l=1}^{L} \phi(a_{l-1}, W_l, b_l, z_l) + \sum_{l=1}^{L-1} \mathbb{I}(h_l(z_l) - \varepsilon_l \leq a_l \leq h_l(z_l) + \varepsilon_l)$$

The penalty term is defined as $\phi(a_{l-1}, W_l, b_l, z_l) = (\rho/2)\|z_l - W_l a_{l-1} - b_l\|_2^2$, where $\rho > 0$ a penalty parameter. $\mathbf{W} = \{W_l\}_{l=1}^{L}$, $\mathbf{b} = \{b_l\}_{l=1}^{L}$, $\mathbf{z} = \{z_l\}_{l=1}^{L}$, $\mathbf{a} = \{a_l\}_{l=1}^{L-1}$. The reason for introducing $\varepsilon_l$ is that it allows us to project the nonlinear constraint to a convex $\varepsilon_l$-ball, thus transforming the nonconvex Problem 1 into the multi-convex Problem 2, which is much easier to solve. Here, a multi-convex problem means this problem is convex with regard to one variable while fixing others. For example, Problem 2 is convex with regard to $\mathbf{z}$ when $\mathbf{W}$, $\mathbf{b}$, and $\mathbf{a}$ are fixed. As $\rho \to \infty$ and $\varepsilon_l \to 0$, Problem 2 approaches Problem 1.

## 2.2 ALTERNATING OPTIMIZATION

We present DLAM algorithm developed to solve Problem 2, shown in Algorithm 1. Lines 4, 5, 7, and 10 update $W_l$, $b_l$, $z_l$ and $a_l$, respectively, and the four relevant subproblems are discussed as below:
**1. Update $W_l$**
The variables $W_l(l = 1, \cdots, L)$ are updated as follows:

$$W_l^{k+1} \leftarrow \arg\min_{W_l} \phi(a_{l-1}^{k+1}, W_l, b_l^k, z_l^k) + \Omega_l(W_l) \tag{1}$$

Because $W_l$ and $a_{l-1}$ are coupled in $\phi(\bullet)$, solving $W_l$ requires an inversion operation of $a_{l-1}^{k+1}$, which is computationally expensive. Motivated by deep learning Alternating Direction Method of Multipliers (dlADMM) (Wang et al. (2019)), we define $P_l^{k+1}(W_l; \theta_l^{k+1})$ as a quadratic approximation of $\phi$ at $W_l^k$, which is mathematically reformulated as follows (Beck & Teboulle (2009)):

$$P_l^{k+1}(W_l; \theta_l^{k+1}) = \phi(a_{l-1}^{k+1}, W_l^k, z_l^k, b_l^k) + <\nabla_{W_l^k}\phi, W_l - W_l^k> + \|\theta_l^{k+1} \circ (W_l - W_l^k)^{\circ 2}\|_1/2$$

where $\theta_l^{k+1} > 0$ is a parameter vector, $\circ$ denotes Hadamard product (the elementwise product), and $a^{\circ b}$ denotes $a$ to the Hadamard power of $b$ and $\|\bullet\|_1$ is the $\ell_1$ norm. $<\bullet, \bullet>$ is a Frobenius inner product. $\nabla_{W_l^k}\phi = \rho(W_l^k a_{l-1}^{k+1} + b_l^k - z_l^k)(a_{l-1}^{k+1})^T (l = 1, \cdots, L)$ is the gradient of $\phi$ with regard to $W_l$ at $W_l^k$. Obviously, $P_l^{k+1}(W_l^k; \theta_l^{k+1}) = \phi(a_{l-1}^{k+1}, W_l^k, b_l^k, z_l^k)$. Rather than minimizing the

---

**Algorithm 1** DLAM Algorithm for Solving Problem 2

---

**Require:** $y, a_0 = x$.
**Ensure:** $a_l, W_l, b_l, z_l (l = 1, \cdots, L)$.
1: Initialize $\rho, k = 0$.
2: **repeat**
3:   **for** $l = 1$ to $L$ **do**
4:     Update $W_l^{k+1}$ using Algorithm 2.
5:     Update $b_l^{k+1}$ in equation 3.
6:     **if** $l = L$ **then**
7:       Update $z_l^{k+1}$ in equation 5.
8:     **else**
9:       Update $z_l^{k+1}$ in equation 4.
10:       Update $a_l^{k+1}$ using Algorithm 3.
11:     **end if**
12:   **end for**
13:   $k \leftarrow k + 1$.
14: **until** convergence.
15: Output $a_l, W_l, b_l, z_l$.

---

original subproblem in equation 1, we instead minimize the following:

$$W_l^{k+1} \leftarrow \arg\min_{W_l} P_l^{k+1}(W_l; \theta_l^{k+1}) + \Omega_l(W_l) \tag{2}$$

For $\Omega_l(W_l)$, common regularization terms like $\ell_1$ or $\ell_2$ regularizations lead to closed-form solutions. As for the choice of $\theta_l^{k+1}$, the backtracking algorithm is shown in Algorithm 2. Specifically, for a given $\theta_l^{k+1}$, we minimize equation 2 to obtain $W_l^{k+1}$ until the condition in Line 3 is satisfied. The time complexity of Algorithm 2 is $O(d^2)$, where $d$ is the dimension of the neurons or features.

---

**Algorithm 2** Backtracking Algorithm to update $W_l^{k+1}$

---

**Require:** $a_{l-1}^{k+1}, W_l^k, b_l^k, z_l^k, \rho$, some constant $\gamma > 1$.
**Ensure:** $\theta_l^{k+1}, W_l^{k+1}$.
1: Initialize $\alpha$.
2: update $\zeta$ in equation 2 where $\theta_l^{k+1} = \alpha$.
3: **while** $\phi(a_{l-1}^{k+1}, \zeta, b_l^k, z_l^k) > P_l^{k+1}(\zeta; \alpha)$ **do**
4:   $\alpha \leftarrow \alpha\gamma$.
5:   update $\zeta$ in equation 2 where $\theta_l^{k+1} = \alpha$.
6: **end while**
7: Output $\theta_l^{k+1} \leftarrow \alpha$.
8: Output $W_l^{k+1} \leftarrow \zeta$.

---

**2. Update $b_l$**
The variables $b_l(l = 1, \cdots, L)$ are updated as follows:

$$b_l^{k+1} \leftarrow \arg\min_{b_l} \phi(a_{l-1}^{k+1}, W_l^{k+1}, b_l, z_l^k).$$

Similarly to updating $W_l$, we define $U_l^{k+1}(b_l; L_b)$ as a quadratic approximation of $\phi$ at $b_l^k$, which is formulated mathematically as follows (Beck & Teboulle (2009)):

$$U_l^{k+1}(b_l; L_b) = \phi(a_{l-1}^{k+1}, W_l^{k+1}, b_l^k, z_l^k) + (\nabla_{b_l^k}\phi)^T(b_l - b_l^k) + (L_b/2)\|b_l - b_l^k\|_2^2.$$

where $L_b \geq \rho$ is a parameter and $\nabla_{b_l^k}\phi = \rho(b_l^k + W_l^{k+1}a_{l-1}^{k+1} - z_l^k)$. Here, $L_b \geq \rho$ is required for the convergence analysis (Beck & Teboulle (2009)). Without loss of generality, we set $L_b = \rho$. We can now solve the following subproblem:

$$b_l^{k+1} \leftarrow \arg\min_{b_l} U_l^{k+1}(b_l; \rho) \tag{3}$$

The solution to equation 3 is: $b_l^{k+1} \leftarrow b_l^k - \nabla_{b_l^k}\phi/\rho$.

**3. Update $z_l$**
The variables $z_l(l = 1, \cdots, L)$ are updated as follows:

$$z_l^{k+1} \leftarrow \arg\min_{z_l} \phi(a_{l-1}^{k+1}, W_l^{k+1}, b_l^{k+1}, z_l) + \mathbb{I}(h_l(z_l) - \varepsilon_l \leq a_l^k \leq h_l(z_l) + \varepsilon_l)(l < L)$$

$$z_L^{k+1} \leftarrow \arg\min_{z_L} \phi(a_{L-1}^{k+1}, W_L^{k+1}, b_L^{k+1}, z_L) + R(z_L; y)$$

As when updating $b_l$, we define $V_l^{k+1}(z_l; L_z)$ as a quadratic approximation of $\phi$ at $z_l^k$, which is formulated mathematically as follows:

$$V_l^{k+1}(z_l; L_z) = \phi(a_{l-1}^{k+1}, W_l^{k+1}, b_l^{k+1}, z_l^k) + (\nabla_{z_l^k}\phi)^T(z_l - z_l^k) + (L_z/2)\|z_l - z_l^k\|_2^2$$

where $L_z \geq \rho$ is a parameter and $\nabla_{z_l^k}\phi = \rho(z_l^k - W_l^{k+1}a_{l-1}^{k+1} - b_l^{k+1})$. Without loss of generality, we set $L_z = \rho$. Obviously, $V_l^{k+1}(z_l^k; \rho) = \phi(a_{l-1}^{k+1}, W_l^{k+1}, b_l^{k+1}, z_l^k)$. Hence, we solve the problems:

$$z_l^{k+1} \leftarrow \arg\min_{z_l} V_l^{k+1}(z_l; \rho) + \mathbb{I}(h_l(z_l) - \varepsilon_l \leq a_l^k \leq h_l(z_l) + \varepsilon_l)(l < L) \tag{4}$$

$$z_L^{k+1} \leftarrow \arg\min_{z_L} V_L^{k+1}(z_L; \rho) + R(z_L; y) \tag{5}$$

As for $z_l(l = 1, \cdots, l - 1)$, the solution is

$$z_l^{k+1} \leftarrow \min(\max(B_1^{k+1}, z_l^k - \nabla\phi_{z_l^k}/\rho), B_2^{k+1}).$$

where $B_1^{k+1}$ and $B_2^{k+1}$ represent the lower bound and the upper bound of the set $\{z_l | h_l(z_l) - \varepsilon_l \leq a_l^k \leq h_l(z_l) + \varepsilon_l\}$. equation 5 is easy to solve using the Fast Iterative Soft Thresholding Algorithm (FISTA) (Beck & Teboulle (2009)).

**4. Update $a_l$**

The variables $a_l(l = 1, \cdots, L - 1)$ are updated as follows:

$$a_l^{k+1} \leftarrow \arg\min_{a_l} \phi(a_l, W_{l+1}^k, b_{l+1}^k, z_{l+1}^k) + \mathbb{I}(h_l(z_l^{k+1}) - \varepsilon_l \leq a_l \leq h_l(z_l^{k+1}) + \varepsilon_l)$$

As when solving $W_l^{k+1}$, the quadratic approximation of $\phi$ at $a_l^k$ is defined as

$$Q_l^{k+1}(a_l; \tau_l^{k+1}) = \phi(a_l^k, W_{l+1}^k, b_{l+1}^k, z_{l+1}^k) + (\nabla_{a_l^k}\phi)^T(a_l - a_l^k) + \|\tau_l^{k+1} \circ (a_l - a_l^k)^{\circ 2}\|_1/2$$

and this allows us to solve the following problem instead:

$$a_l^{k+1} \leftarrow \arg\min_{a_l} Q_l^{k+1}(a_l; \tau_l^{k+1}) + \mathbb{I}(h_l(z_l^{k+1}) - \varepsilon_l \leq a_l \leq h_l(z_l^{k+1}) + \varepsilon_l) \tag{6}$$

where $\tau_l^{k+1} > 0$ is a parameter vector. $\nabla_{a_l^k}\phi = \rho(W_{l+1}^k)^T(W_{l+1}^k a_l^k + b_{l+1}^k - z_{l+1}^k)(l = 1, \cdots, L-1)$ is the gradient of $\phi$ with regard to $a_l$ at $a_l^k$. Obviously, $Q_l^{k+1}(a_l^k; \tau_l^{k+1}) = \phi(a_l^k, W_{l+1}^k, b_{l+1}^k, z_{l+1}^k)$. Because $Q_l^{k+1}(a_l; \tau_l^{k+1})$ is a quadratic function with respect to $a_l$, the solution can be obtained by $a_l^{k+1} \leftarrow a_l^k - \nabla_{a_l^k}\phi/\tau_l^{k+1}$ given a suitable $\tau_l^{k+1}$. Now the main focus is how to choose $\tau_l^{k+1}$. Similar to Algorithm 2, the backtracking algorithm for finding a suitable $\tau_l^{k+1}$ is shown in Algorithm 3. The time complexity of Algorithm 3 is $O(d^2)$, where $d$ is the dimension of neurons or features.

---

**Algorithm 3** Backtracking Algorithm to update $a_l^{k+1}$

---

**Require:** $a_l^k, W_{l+1}^k, z_l^{k+1}, z_{l+1}^k, b_{l+1}^k, \rho$, some constant $\eta > 1$.
**Ensure:** $\tau_l^{k+1}, a_l^{k+1}$.
1: Pick up $t$ such that $\beta = a_l^k - \nabla_{a_l^k}\phi/t$ and $h_l(z_l^{k+1}) - \varepsilon_l \leq \beta \leq h_l(z_l^{k+1}) + \varepsilon_l$.
2: **while** $\phi(\beta, W_{l+1}^k, z_{l+1}^k, b_{l+1}^k) > Q_l^{k+1}(\beta; t)$ **do**
3:    $t \leftarrow t\eta$.
4:    $\beta \leftarrow a_l^k - \nabla\phi_{a_l^k}/t$.
5: **end while**
6: Output $\tau_l^{k+1} \leftarrow t$.
7: Output $a_l^{k+1} \leftarrow \beta$.

---

## 3 CONVERGENCE ANALYSIS

In this section, we present the main convergence analyses for the DLAM algorithm. Specifically, Section 3.1 introduces the assumption necessary to guarantee convergence. The main convergence properties of the new DLAM algorithm are presented in Section 3.2. Due to space limit, the discussion on the convergence conditions of the DLAM algorithm is in Section G in the supplementary materials.

### 3.1 ASSUMPTION

Firstly, we make the following assumption:

**Assumption 1** (Quasilinearity). *Activation functions $h_l(z_l)(l = 1, \cdots, n)$ are quasilinear functions.*

The definition of Quasilinearity is given in the supplementary material. Assumption 1 is a mild condition to ensure that the nonlinear constraint $a_l = h_l(z_l)$ in Problem 1 is projected in a convex

set, and it allows for nonsmooth functions. Fortunately, most of the widely used nonlinear activation functions, including tanh (Zamanlooy & Mirhassani (2014)), smooth sigmoid (Glorot & Bengio (2010)), and the rectified linear unit (Relu) (Maas et al. (2013)) that are quasilinear.

Notice that no assumption is needed to imposed on the risk function $R(z_L; y)$ and the regularization term $\Omega_l(W_l)$. Therefore, they can be either smooth or nonsmooth: $R(z_L; y)$ can be a common least square loss or cross entropy loss; $\Omega_l(W_l)$ can be a either $\ell_1$ or $\ell_2$ regularization term. They fit neatly into our framework and incorporate several important theoretical properties.

### 3.2 KEY CONVERGENCE PROPERTIES

We introduce several important convergence properties possessed by the DLAM algorithm in this section. If Assumption 1 holds, then Lemmas 1-3 stated below are satisfied. These three lemmas are proven to be possessed by the DLAM algorithm, and are key for demonstrating the theoretical merits of DLAM; the proofs of them are provided in the supplementary materials. Finally, the global convergence and convergence rate of the DLAM are proved based on Lemmas 1-3 stated as follows:

**Lemma 1** (Sufficient Descent). *For any $\rho > 0$ and $\varepsilon_l > 0$, we have*

$$F(\boldsymbol{W}^k, \boldsymbol{b}^k, \boldsymbol{z}^k, \boldsymbol{a}^k) - F(\boldsymbol{W}^{k+1}, \boldsymbol{b}^{k+1}, \boldsymbol{z}^{k+1}, \boldsymbol{a}^{k+1}) \geq \sum_{l=1}^{L} \|\theta_l^{k+1} \circ (W_l^{k+1} - W_l^k)^{\circ 2}\|_1 / 2 +$$

$$(\rho/2)\sum_{l=1}^{L} \|b_l^{k+1} - b_l^k\|_2^2 + (\rho/2)\sum_{l=1}^{L} \|z_l^{k+1} - z_l^k\|_2^2 + \sum_{l=1}^{L-1} \|\tau_l^{k+1} \circ (a_l^{k+1} - a_l^k)^{\circ 2}\|_1 / 2 \quad (7)$$

Lemma 1 depicts the monotonic decrease of the objective value during iterations. The proof of Lemma 1 requires Assumption 1 and is detailed in the supplementary materials.

**Lemma 2** (Convergent Sequence). *(a). $(\boldsymbol{W}^k, \boldsymbol{b}^k, \boldsymbol{z}^k, \boldsymbol{a}^k)$ and $F(\boldsymbol{W}^k, \boldsymbol{b}^k, \boldsymbol{z}^k, \boldsymbol{a}^k)$ are convergent. That is, as $k \to \infty$, $(\boldsymbol{W}^k, \boldsymbol{b}^k, \boldsymbol{z}^k, \boldsymbol{a}^k) \to (\boldsymbol{W}^*, \boldsymbol{b}^*, \boldsymbol{z}^*, \boldsymbol{a}^*)$ and $F(\boldsymbol{W}^k, \boldsymbol{b}^k, \boldsymbol{z}^k, \boldsymbol{a}^k) \to F(\boldsymbol{W}^*, \boldsymbol{b}^*, \boldsymbol{z}^*, \boldsymbol{a}^*)$.*
*(b). $(\boldsymbol{W}^k, \boldsymbol{b}^k, \boldsymbol{z}^k, \boldsymbol{a}^k)$ is bounded. That is, there exist scalars $M_{\boldsymbol{W}}, M_{\boldsymbol{b}}, M_{\boldsymbol{z}}$ and $M_{\boldsymbol{a}}$ such that $\|\boldsymbol{W}^k\| \leq M_{\boldsymbol{W}}$, $\|\boldsymbol{b}^k\| \leq M_{\boldsymbol{b}}$, $\|\boldsymbol{z}^k\| \leq M_{\boldsymbol{z}}$ and $\|\boldsymbol{a}^k\| \leq M_{\boldsymbol{a}}$.*

Lemma 2 guarantees that the variable $(\mathbf{W}^k, \mathbf{b}^k, \mathbf{z}^k, \mathbf{a}^k)$ is convergent and bounded. The proof of Lemma 2 requires Lemma 1 and can be found in the supplementary materials.

**Lemma 3** (Subgradient Bound). *There exist $C = \max(\rho M_{\boldsymbol{a}}, \rho M_{\boldsymbol{a}}^2 + \|\theta_1^{k+1}\|, \rho M_{\boldsymbol{a}}^2 + \|\theta_2^{k+1}\|, \cdots, \rho M_{\boldsymbol{a}}^2 + \|\theta_L^{k+1}\|)$, some $g_1^{k+1} \in \partial_{\boldsymbol{W}^{k+1}} F$ and $g_2^{k+1} = \nabla_{\boldsymbol{b}^{k+1}} F$ such that*

$$\|g_1^{k+1}\| \leq C(\|\boldsymbol{W}^{k+1} - \boldsymbol{W}^k\| + \|\boldsymbol{b}^{k+1} - \boldsymbol{b}^k\| + \|\boldsymbol{z}^{k+1} - \boldsymbol{z}^k\|), \quad \|g_2^{k+1}\| = \rho\|\boldsymbol{z}^{k+1} - \boldsymbol{z}^k\|$$

Lemma 3 ensures that the subgradient of the objective function is bounded by variables. The proof of Lemma 3 requires Lemma 2 and the proof process is elaborated in the supplementary materials. We will now move on to present the global convergence of the DLAM algorithm using the following two theorems. The first theorem presents the global convergence of the DLAM algorithm.

**Theorem 1** (Convergence to the Critical Point). *For $(\boldsymbol{W}, \boldsymbol{b})$ in Problem 2, starting from any $(\boldsymbol{W}^0, \boldsymbol{b}^0)$, Algorithm 1 converges to a critical point $(\boldsymbol{W}^*, \boldsymbol{b}^*)$. That is, $0 \in \partial_{\boldsymbol{W}^*} F$ and $0 \in \partial_{\boldsymbol{b}^*} F$.*

*Proof.* By Lemma 2, $(\mathbf{W}^k, \mathbf{b}^k)$ is convergent to $(\mathbf{W}^*, \mathbf{b}^*)$. From Lemma 3, there exist $g_1^{k+1} \in \partial_{\mathbf{W}^{k+1}} F$ and $g_2^{k+1} \in \partial_{\mathbf{b}^{k+1}} F$ such that $\|g_1^{k+1}\| \to 0$ and $\|g_2^{k+1}\| \to 0$ as $k \to \infty$. we have $0 \in \partial_{\mathbf{W}^*} F$ and $0 \in \partial_{\mathbf{b}^*} F$. In other words, $(\mathbf{W}^*, \mathbf{b}^*)$ is a critical point of $F$. $\quad\square$

Theorem 1 shows that our proposed DLAM algorithm converges to a critical point globally no matter what $\rho$ and $\varepsilon_l$ are chosen. This ensures that our DLAM algorithm is parameter-restriction free, so the choice of hyperparameters has no effect on its convergence.

The next theorem shows that the convergence rate of DLAM is $o(1/k)$, which is shown as follows:

**Theorem 2** (Convergence Rate). *For $(\boldsymbol{W}^k, \boldsymbol{b}^k, \boldsymbol{z}^k, \boldsymbol{a}^k)$, define $c_k = \min_{0 \leq i \leq k} (\sum_{l=1}^{L} \|\theta_l^{i+1} \circ (W_l^{i+1} - W_l^i)^{\circ 2}\|_1 / 2 + (\rho/2)\sum_{l=1}^{L} \|b_l^{i+1} - b_l^i\|_2^2 + (\rho/2)\sum_{l=1}^{L} \|z_l^{i+1} - z_l^i\|_2^2 + \sum_{l=1}^{L-1} \|\tau_l^{i+1} \circ (a_l^{i+1} - a_l^i)^{\circ 2}\|_1 / 2)$, which reflects the convergence rate of Algorithm 1, then the convergence rate of $c_k$ is $o(1/k)$.*

*Proof.* The proof of this theorem is in the supplementary materials. $\quad\square$

## 4 EXPERIMENTS

The DLAM algorithm is evaluated by several benchmark datasets. Effectiveness, efficiency and convergence properties of DLAM are compared with state-of-the-art methods. All experiments were conducted on 64-bit Ubuntu16.04 LTS with Intel(R) Xeon processor and GTX1080Ti GPU.

### 4.1 EXPERIMENT SETUP

#### 4.1.1 DATASET

In this experiment, two benchmark datasets were used for comparison: MNIST (LeCun et al. (1998)) and Fashion MNIST (Xiao et al. (2017)). The MNIST dataset has ten classes of handwritten-digit images, which was firstly introduced by Lecun et al. in 1998 (LeCun et al. (1998)). It contains 55,000 training samples and 10,000 test samples with 196 features each, which is provided by the Keras library (Chollet (2017)). Unlike the MNIST dataset, the Fashion MNIST dataset has ten classes of assortment images on the website of Zalando, which is Europes largest online fashion platform (Xiao et al. (2017)). The Fashion-MNIST dataset consists of 60,000 training samples and 10,000 test samples with 784 features each.

#### 4.1.2 EXPERIMENT SETTINGS

We set up two different architectures of multi-layer neural networks in the experiment. Two network structures contained two hidden layers with $100$ and $500$ hidden units each, respectively. The rectified linear unit (Relu) was used for the activation function for both network structures. The loss function was set as the deterministic cross-entropy loss. $\rho$ was set to $10^{-4}$. $\varepsilon$ was initialized as $10$ and updated adaptively as follows: if $R(z_L^k; y) > 10\varepsilon^k$, $\varepsilon^{k+1} = \max(2\varepsilon^k, 1)$; if $\varepsilon^k > 10R(z_L^k; y)$, $\varepsilon^{k+1} = \min(\varepsilon^k/2, 0.01)$, which balances between the loss function $R(z_L; y)$ and $\varepsilon$. The number of iteration was set to $150$. In the experiment, one iteration means one epoch.

#### 4.1.3 COMPARISON METHODS

Since this paper focuses on fully-connected deep neural networks, SGD and its variants and ADMM are state-of-the-art methods and hence were served as comparison methods. For SGD-based methods, the full batch dataset is used for training models. All parameters were chosen by the accuracy of the training dataset. The baselines are: **1) Stochastic Gradient Descent (SGD) (Bottou (2010))**. The SGD and its variants are the most popular deep learning optimizers, whose convergence has been studied extensively in the literature. **2) Adaptive gradient algorithm (Adagrad) (Duchi et al. (2011))**. Adagrad is an improved version of SGD: rather than fixing the learning rate during iteration, it adapts the learning rate to the hyperparameter. **3) Adaptive learning rate method (Adadelta) (Zeiler (2012))**. As an improved version of the Adagrad, the Adadelta is proposed to overcome the sensitivity to hyperparameter selection. **4) Alternating Direction Method of Multipliers (ADMM) (Taylor et al. (2016))**. ADMM is a powerful convex optimization method because it can split an objective function into a series of subproblems, which are coordinated to get global solutions. It is scalable to large-scale datasets and supports parallel computations.

### 4.2 EXPERIMENTAL RESULTS

In this section, experimental results of DLAM algorithm are analyzed against comparison methods.

#### 4.2.1 CONVERGENCE

First, we show that our proposed DLAM algorithm converges for both the MNIST dataset and the Fashion MNIST dataset. The convergence of DLAM algorithm is shown in Figure 1. The X axis and Y axis denote the number of iterations and the logarithm of objective value, respectively. Overall, the objective value decreased monotonically during iteration whatever network structures and datasets we choose. Specifically, the objective value dropped tremendously at the early stage and then converged smoothly towards the critical point of the problem. We also found that the objective value for the Fashion MNIST dataset decreased more quickly than that for the MNIST dataset.

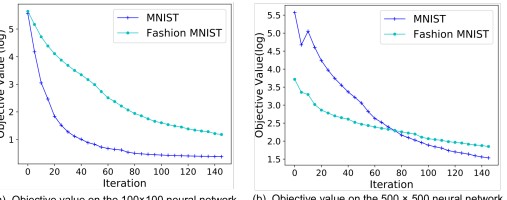

(a). Objective value on the 100×100 neural network    (b). Objective value on the 500 × 500 neural network

Figure 1: Convergence curves of DLAM algorithm on MNIST and Fashion MNIST datasets for two neural network structures: DLAM algorithm converged.

#### 4.2.2 PERFORMANCE

Figure 2 and Figure 3 show the curves of the training accuracy and test accuracy of our proposed DLAM algorithm and baselines, respectively. Overall, both the training accuracy

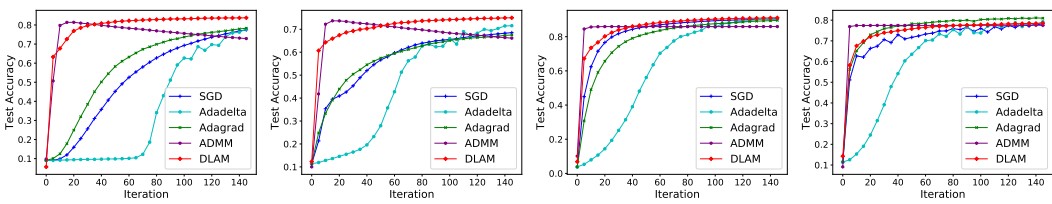

(a) Size of 100×100 on MNIST    (b) Size of 100×100 on Fashion MNIST    (c) Size of 500×500 on MNIST    (d) Size of 500×500 on Fashion MNIST

Figure 2: Training accuracy of all methods for the MNIST and Fashion MNIST datasets on two neural network structures: DLAM algorithm performed competitively.

(a) Size of 100×100 on MNIST    (b) Size of 100×100 on Fashion MNIST    (c) Size of 500×500 on MNIST    (d) Size of 500×500 on Fashion MNIST

Figure 3: Test accuracy of all methods for the MNIST and Fashion MNIST datasets on two neural network structures: DLAM algorithm performed competitively.

and the test accuracy of our proposed DLAM outperformed all baselines for the MNIST dataset, while those of our proposed DLAM algortihm performed competitively for the Fashion MNIST dataset. Specifically, the curves of our DLAM algorithm soared to $0.7$ at the early stage, and then raised steadily towards to $0.8$ or more. The curves of the SGD-related methods, SGD, Adadelta, and Adagrad, moved more slowly than our proposed DLAM algorithm. The curves of the ADMM also rocketed to around $0.8$, but decreased slightly later on.

### 4.2.3 EFFICIENCY

In this subsection, the relationship between running time per iteration of our proposed DLAM algorithm and two potential factors, namely, the value of $\rho$, the size of training sample was explored. The running time was calculated by the

| MNIST dataset: From 11,000 to 55,000 training samples | | | | | |
|---|---|---|---|---|---|
| size
$\rho$ | 11000 | 22000 | 33000 | 44000 | 55000 |
| 0.0001 | 0.1692 | 0.3216 | 0.5010 | 0.7164 | 0.9413 |
| 0.001 | 0.2061 | 0.4328 | 0.6951 | 0.9792 | 1.2442 |
| 0.01 | 0.3334 | 0.6516 | 1.0277 | 1.3956 | 1.7783 |
| 0.1 | 0.4795 | 0.9428 | 1.4524 | 1.959 | 2.4410 |
| 1 | 0.7684 | 1.4810 | 2.2626 | 3.0299 | 3.7504 |
| Fashion MNIST dataset: From 12,000 to 60,000 training samples | | | | | |
| size
$\rho$ | 12,000 | 24,000 | 36,000 | 48,000 | 60,000 |
| 0.0001 | 0.2500 | 0.5081 | 0.8492 | 1.1911 | 1.5092 |
| 0.001 | 0.2980 | 0.5980 | 0.9595 | 1.3265 | 1.6744 |
| 0.01 | 0.4199 | 0.8028 | 1.2787 | 1.7535 | 2.2025 |
| 0.1 | 0.5758 | 1.0928 | 1.7230 | 2.3261 | 2.9234 |
| 1 | 0.8795 | 1.6464 | 2.5580 | 3.4492 | 4.2902 |

Table 2: The relation between running time per iteration (in second) and size of training samples as well as value of $\rho$: generally, the running time increased as the training sample and the value of $\rho$ became larger.

average of 150 iterations. The computational result for the MNIST dataset and Fashion MNIST dataset on the $100 \times 100$ neural network is shown in Table 2. The number of training samples of the MNIST dataset ranged from 11,000 to 55,000, with an increase of 11,000 each time, whereas The number of training samples of the Fashion MNIST dataset ranged from 12,000 to 60,000, with an increase of 12,000 each time. The value of $\rho$ ranged from 0.0001 to 1, with multiplying by 10 each time. Generally, the running time increased as the training sample and the value of $\rho$ became larger.

## 5 CONCLUSION

Even though stochastic gradient descent (SGD) is a popular method to train deep neural networks, alternating minimization methods have attracted increasing attention from a great deal of researchers recently as they have several advantages including solid theoretical guarantees and avoiding gradient vanishing problems. In this paper, we propose a novel formulation of the original deep neural network problem and a novel Deep Learning Alternating Minimization (DLAM) algorithm. Specifically, the nonlinear constraint is projected into a convex set so that all subproblems are solvable. At the same time, the quadratic approximation technique and the backtracking algorithm are applied to boost up scalability. Furthermore, several mild assumptions are established to prove the global convergence of our DLAM algorithm. Experiments on real-world datasets demonstrate the convergence, effectiveness, and efficiency of our DLAM algorithm.

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

## Appendix

## A  DEFINITIONS

First, the definition of Fréchet subdifferential is shown as follows (Rockafellar & Wets (2009)):

**Definition 1** (Fréchet Subdifferential). *For each $x_1 \in dom(u_1)$, the Fréchet subdifferential of $u_1$ at $x_1$, which is denoted as $\hat{\partial}u_1(x_1)$, is the set of vectors $v$, which satisfy*

$$\lim_{x_2 \neq x_1} \inf_{x_2 \to x_1} (u_1(x_2) - u_1(x_1) - v^T(x_2 - x_1))/\|x_2 - x_1\| \geq 0.$$

*The vector $v \in \hat{\partial}u_1(x_1)$ is a Fréchet subgradient.*

Then the definition of the limiting subdifferential, which is based on Fréchet subdifferential, is given in the following (Rockafellar & Wets (2009)):

**Definition 2** (Limiting Subdifferential). *For each $x \in dom(u_2)$, the limiting subdifferential (or subdifferential) of $u_2$ at $x$ is*

$$\partial u_2(x) = \{v_1 | \exists\, x^k \to x, s.t.\ u_2(x^k) \to u_2(x),$$
$$v^k \in \hat{\partial}u_2(x^k), v^k \to v\}$$

*where $x^k$ is a sequence whose limit is $x$ and the limit of $u_2(x^k)$ is $u_2(x)$, $v^k$ is a sequence, which is a Fréchet subgradient of $u_2$ at $x^k$ and whose limit is $v$. The vector $v \in \partial u_2(x)$ is a limiting subgradient.*

Specifically, when $u_2$ is convex, its limiting subdifferential is reduced to regular subdifferential (Rockafellar & Wets (2009)); The limiting subdifferential is used to prove the global convergence of the DLAM in the following convergence analysis. Without loss of generality, $\partial R$ and $\partial \Omega_l (l = 1, \cdots, n)$ are nonempty, and the limiting subdifferential of $F$ defined in Problem 2 is (Xu & Yin (2013)):

$$\partial F(\mathbf{W}, \mathbf{b}, \mathbf{z}, \mathbf{a}) = \partial_{\mathbf{W}} F \times \nabla_{\mathbf{b}} F \times \partial_{\mathbf{z}} F \times \partial_{\mathbf{a}} F$$

where $\times$ means the Cartesian product.

Next, recall the definition of quasilinearity (Boyd & Vandenberghe (2004)):

**Definition 3.** *A function $f(x)$ is quasiconvex if for any sublevel set $S_\alpha(f) = \{x | f(x) \leq \alpha\}$ is a convex set. Likewise, A function $f(x)$ is quasiconcave if for any superlevel set $S_\alpha(f) = \{x | f(x) \geq \alpha\}$ is a convex set. A function $f(x)$ is quasilinear if it is both quasiconvex and quasiconcave.*

## B  PRELIMINARY LEMMAS

In this section, we give preliminary lemmas which are useful for the proofs of three properties. The proofs of Lemmas 4 and 5 both require Assumption 1. The proof of Lemma 6 requires Lemmas 4 and 5. To simplify the notation, $\mathbf{W}_{\leq l}^{k+1} = \{\{W_i^{k+1}\}_{i=1}^l, \{W_i^k\}_{i=l+1}^L\}$, $\mathbf{b}_{\leq l}^{k+1} = \{\{b_i^{k+1}\}_{i=1}^l, \{b_i^k\}_{i=l+1}^L\}$, $\mathbf{z}_{\leq l}^{k+1} = \{\{z_i^{k+1}\}_{i=1}^l, \{z_i^k\}_{i=l+1}^L\}$ and $\mathbf{a}_{\leq l}^{k+1} = \{\{a_i^{k+1}\}_{i=1}^l, \{a_i^k\}_{i=l+1}^{L-1}\}$.

**Lemma 4.** *equation 2 holds if and only if there exists $s \in \partial \Omega_l(W_l^{k+1})$, the subgradient of $\Omega_l(W_l^{k+1})$ such that*

$$\nabla_{W_l^k}\phi + \theta_l^{k+1} \circ (W_l^{k+1} - W_l^k) + s = 0$$

*Likewise, equation 4 holds if and only if there exists $r \in \partial \mathbb{I}(h_l(z_l^{k+1}) - \varepsilon_l \leq a_l^k \leq h_l(z_l^{k+1}) + \varepsilon_l)$ such that*

$$\nabla_{z_l^k}\phi + \rho(z_l^{k+1} - z_l^k) + r = 0$$

*equation 5 holds if and only if there exists $u \in \partial R(z_L^{k+1}; y)$ such that*

$$\nabla_{z_L^k}\phi + \rho(z_L^{k+1} - z_L^k) + u = 0$$

*equation 6 holds if and only if there exists $v \in \partial \mathbb{I}(h_l(z_l^{k+1}) - \varepsilon_l \leq a_l^{k+1} \leq h_l(z_l^{k+1}) + \varepsilon_l)$ such that*

$$\nabla_{a_l^k}\phi + \rho(a_l^{k+1} - a_l^k) + v = 0$$

*Proof.*  These can be obtained by directly applying the optimality conditions of equation 2, equation 4, equation 5 and equation 6, respectively. □

**Lemma 5.** *For equation 4, equation 5 and equation 3, if $L_b \geq \rho$ and $L_z \geq \rho$, then the following inequalities hold:*

$$U_l^{k+1}(b_l^{k+1}; L_b) \geq \phi(a_{l-1}^{k+1}, W_l^{k+1}, b_l^{k+1}, z_l^k) \tag{8}$$

$$V_l^{k+1}(z_l^{k+1}; L_z) \geq \phi(a_{l-1}^{k+1}, W_l^{k+1}, b_l^{k+1}, z_l^{k+1}) \tag{9}$$

*Proof.* Because $\phi(a_{l-1}, W_l, b_l, z_l)$ is differentiable continuous with respect to $b_l$ and $z_l$ with Lipschitz coefficient $\rho$ (the definition of Lipschitz differentiablity can be found in (Beck & Teboulle (2009))), we directly apply Lemma 2.1 in (Beck & Teboulle (2009)) to $\phi$ to obtain equation 8 and equation 9, respectively. □

**Lemma 6.** *It holds that for $\forall k \in \mathbb{N}$ and $l = 1, 2, \cdots, L$,*

$$F(\boldsymbol{W}_{\leq l-1}^{k+1}, \boldsymbol{b}_{\leq l-1}^{k+1}, \boldsymbol{z}_{\leq l-1}^{k+1}, \boldsymbol{a}_{\leq l-1}^{k+1}) - F(\boldsymbol{W}_{\leq l}^{k+1}, \boldsymbol{b}_{\leq l-1}^{k+1}, \boldsymbol{z}_{\leq l-1}^{k+1}, \boldsymbol{a}_{\leq l-1}^{k+1})$$
$$\geq \|\theta_l^{k+1} \circ (W_l^{k+1} - W_l^k)^{\circ 2}\|_1/2. \tag{10}$$

$$F(\boldsymbol{W}_{\leq l}^{k+1}, \boldsymbol{b}_{\leq l-1}^{k+1}, \boldsymbol{z}_{\leq l-1}^{k+1}, \boldsymbol{a}_{\leq l-1}^{k+1}) - F(\boldsymbol{W}_{\leq l}^{k+1}, \boldsymbol{b}_{\leq l}^{k+1}, \boldsymbol{z}_{\leq l-1}^{k+1}, \boldsymbol{a}_{\leq l-1}^{k+1})$$
$$\geq (\rho/2)\|b_l^{k+1} - b_l^k\|_2^2. \tag{11}$$

$$F(\boldsymbol{W}_{\leq l}^{k+1}, \boldsymbol{b}_{\leq l}^{k+1}, \boldsymbol{z}_{\leq l-1}^{k+1}, \boldsymbol{a}_{\leq l-1}^{k+1}) - F(\boldsymbol{W}_{\leq l}^{k+1}, \boldsymbol{b}_{\leq l}^{k+1}, \boldsymbol{z}_{\leq l}^{k+1}, \boldsymbol{a}_{\leq l-1}^{k+1})$$
$$\geq (\rho/2)\|z_l^{k+1} - z_l^k\|_2^2. \tag{12}$$

$$F(\boldsymbol{W}_{\leq l}^{k+1}, \boldsymbol{b}_{\leq l}^{k+1}, \boldsymbol{z}_{\leq l}^{k+1}, \boldsymbol{a}_{\leq l-1}^{k+1}) - F(\boldsymbol{W}_{\leq l}^{k+1}, \boldsymbol{b}_{\leq l}^{k+1}, \boldsymbol{z}_{\leq l}^{k+1}, \boldsymbol{a}_{\leq l}^{k+1})$$
$$\geq \|\tau_l^{k+1} \circ (a_l^{k+1} - a_l^k)^{\circ 2}\|_1/2. \tag{13}$$

*Proof.* Essentially, all inequalities can be obtained by applying optimality conditions of updating $W_l^{k+1}$, $b_l^{k+1}$, $z_l^{k+1}$ and $a_l^{k+1}$, respectively. We only prove equation 10 and equation 12 since equation 13 and equation 11 follow the same routine of equation 10 and equation 12, respectively.
Firstly, we focus on proving equation 10. The stopping criterion of Algorithm 2 shows that

$$\phi(a_{l-1}^{k+1}, W_l^{k+1}, b_l^k, z_l^k) \leq P_l^{k+1}(W_l^{k+1}; \theta_l^{k+1}). \tag{14}$$

Because $\Omega_{W_l}(W_l)$ is convex, according to the definition of subgradient, we have

$$\Omega_l(W_l^k) \geq \Omega_l(W_l^{k+1}) + s^T(W_l^k - W_l^{k+1}) \tag{15}$$

where $s$ is defined in the premise of Lemma 4. Therefore, we have

$$F(\boldsymbol{W}_{\leq l-1}^{k+1}, \boldsymbol{b}_{\leq l-1}^{k+1}, \boldsymbol{z}_{\leq l-1}^{k+1}, \boldsymbol{a}_{\leq l-1}^{k+1}) - F(\boldsymbol{W}_{\leq l}^{k+1}, \boldsymbol{b}_{\leq l-1}^{k+1}, \boldsymbol{z}_{\leq l-1}^{k+1}, \boldsymbol{a}_{\leq l-1}^{k+1})$$
$$= \phi(a_{l-1}^{k+1}, W_l^k, b_l^k, z_l^k) + \Omega_l(W_l^k) - \phi(a_{l-1}^{k+1}, W_l^{k+1}, b_l^k, z_l^k)$$
$$- \Omega_l(W_l^{k+1}) \text{ (Definition of } F \text{ in Problem 2)}$$
$$\geq \Omega_l(W_l^k) - \Omega_l(W_l^{k+1}) - (\nabla_{W_l^k}\phi)^T(W_l^{k+1} - W_l^k)$$
$$- \|\theta_l^{k+1} \circ (W_l^{k+1} - W_l^k)^{\circ 2}\|_1/2(\text{ equation 14})$$
$$\geq s^T(W_l^k - W_l^{k+1}) - (\nabla_{W_l^k}\phi)^T(W_l^{k+1} - W_l^k) -$$
$$\|\theta_l^{k+1} \circ (W_l^{k+1} - W_l^k)^{\circ 2}\|_1/2(\text{ equation 15})$$
$$= (s + \nabla\phi_{W_l^k}^T)(W_l^k - W_l^{k+1}) - \|\theta_l^{k+1} \circ (W_l^{k+1} - W_l^k)^{\circ 2}\|_1/2$$
$$= \|\theta_l^{k+1} \circ (W_l^{k+1} - W_l^k)^{\circ 2}\|_1/2 \text{ (Lemma 4).}$$

Secondly, we focus on proving equation 12. For $l < L$, because $\mathbb{I}(h_l(z_l) - \varepsilon_l \leq a_l^k \leq h_l(z_l) + \varepsilon_l)$ is convex with regard to $z_l$, according to the definition of subgradient, we have

$$\mathbb{I}(h_l(z_l^k) - \varepsilon_l \leq a_l^k \leq h_l(z_l^k) + \varepsilon_l)$$
$$\geq \mathbb{I}(h_l(z_l^{k+1}) - \varepsilon_l \leq a_l^k \leq h_l(z_l^{k+1}) + \varepsilon_l) + r^T(z_l^k - z_l^{k+1}) \tag{16}$$

where $r$ is defined in Lemma 4.

$$F(\mathbf{W}_{\leq l}^{k+1}, \mathbf{b}_{\leq l}^{k+1}, \mathbf{z}_{\leq l-1}^{k+1}, \mathbf{a}_{\leq l-1}^{k+1}) - F(\mathbf{W}_{\leq l}^{k+1}, \mathbf{b}_{\leq l}^{k+1}, \mathbf{z}_{\leq l}^{k+1}, \mathbf{a}_{\leq l-1}^{k+1})$$

$$= \phi(a_{l-1}^{k+1}, W_l^{k+1}, b_l^{k+1}, z_l^k) + \mathbb{I}(h_l(z_l^k) - \varepsilon_l \leq a_l^k \leq h_l(z_l^k) + \varepsilon_l)$$

$$- \phi(a_{l-1}^{k+1}, W_l^{k+1}, b_l^{k+1}, z_l^{k+1}) - \mathbb{I}(h_l(z_l^{k+1}) - \varepsilon_l \leq a_l^k \leq h_l(z_l^{k+1}) + \varepsilon_l)$$

(Definition of $F$ in Problem 2)

$$\geq -(\nabla_{z_l^k} \phi)^T (z_l^{k+1} - z_l^k) - (\rho/2)\|z_l^{k+1} - z_l^k\|_2^2$$

$$+ \mathbb{I}(h_l(z_l^k) - \varepsilon_l \leq a_l^k \leq h_l(z_l^k) + \varepsilon_l)$$

$$- \mathbb{I}(h_l(z_l^{k+1}) - \varepsilon_l \leq a_l^k \leq h_l(z_l^{k+1}) + \varepsilon_l)(\text{ equation } 9)$$

$$\geq -(\nabla_{z_l^k} \phi)^T (z_l^{k+1} - z_l^k) - (\rho/2)\|z_l^{k+1} - z_l^k\|_2^2$$

$$+ r^T (z_l^k - z_l^{k+1})(\text{ equation } 16)$$

$$= -(\nabla_{z_l^k} \phi)^T (z_l^{k+1} - z_l^k) - (\rho/2)\|z_l^{k+1} - z_l^k\|_2^2$$

$$+ (\nabla_{z_l^k} \phi + \rho(z_l^{k+1} - z_l^k))^T (z_l^{k+1} - z_l^k) \text{ (Lemma 4)}$$

$$= (\rho/2)\|z_l^{k+1} - z_l^k\|_2^2.$$

For $z_L$, the same routine applies. $\qquad \square$

## C    PROOF OF LEMMA 1

*Proof.* This can be obtained by adding equation 10, equation 11 and equation 12 from $l = 1$ to $l = L$ and equation 13 from $l = 1$ to $l = L - 1$. $\qquad \square$

## D    PROOF OF LEMMA 2

*Proof.* (a). By Lemma 1, $F(\mathbf{W}^k, \mathbf{b}^k, \mathbf{z}^k, \mathbf{a}^k)$ is non-increasing. Also, $F(\mathbf{W}^k, \mathbf{b}^k, \mathbf{z}^k, \mathbf{a}^k) \geq 0$. According to the monotonic sequence theorem, $F(\mathbf{W}^k, \mathbf{b}^k, \mathbf{z}^k, \mathbf{a}^k)$ is convergent.

Next, we take the limit of Inequality 7 to obtain

$$\lim_{k \to \infty} (F(\mathbf{W}^k, \mathbf{b}^k, \mathbf{z}^k, \mathbf{a}^k) - F(\mathbf{W}^{k+1}, \mathbf{b}^{k+1}, \mathbf{z}^{k+1}, \mathbf{a}^{k+1}))$$

$$\geq \lim_{k \to \infty} \sum_{l=1}^{L} \|\theta_l^{k+1} \circ (W_l^{k+1} - W_l^k)^{\circ 2}\|_1/2$$

$$+ (\rho/2)\sum_{l=1}^{L} \|b_l^{k+1} - b_l^k\|_2^2 + (\rho/2)\sum_{l=1}^{L} \|z_l^{k+1} - z_l^k\|_2^2$$

$$+ \sum_{l=1}^{L-1} \|\tau_l^{k+1} \circ (a_l^{k+1} - a_l^k)^{\circ 2}\|_1/2$$

$$\geq 0$$

Because $F(\mathbf{W}^k, \mathbf{b}^k, \mathbf{z}^k, \mathbf{a}^k)$ is convergent, $\lim_{k \to \infty}(F(\mathbf{W}^k, \mathbf{b}^k, \mathbf{z}^k, \mathbf{a}^k) - F(\mathbf{W}^{k+1}, \mathbf{b}^{k+1}, \mathbf{z}^{k+1}, \mathbf{a}^{k+1})) = 0$. Therefore, we have

$$\lim_{k \to \infty} \sum_{l=1}^{L} \|\theta_l^{k+1} \circ (W_l^{k+1} - W_l^k)^{\circ 2}\|_1/2$$

$$+ (\rho/2)\sum_{l=1}^{L} \|b_l^{k+1} - b_l^k\|_2^2 + (\rho/2)\sum_{l=1}^{L} \|z_l^{k+1} - z_l^k\|_2^2$$

$$+ \sum_{l=1}^{L-1} \|\tau_l^{k+1} \circ (a_l^{k+1} - a_l^k)^{\circ 2}\|_1/2 = 0$$

Since $\theta_l^{k+1}, \tau_l^{k+1} > 0$, we obtain as $k \to \infty$, $\|W_l^{k+1} - W_l^k\| \to 0$, $\|b_l^{k+1} - b_l^k\| \to 0$, $\|z_l^{k+1} - z_l^k\| \to 0$, and $\|a_l^{k+1} - a_l^k\| \to 0$. This shows that $(\mathbf{W}^k, \mathbf{b}^k, \mathbf{z}^k, \mathbf{a}^k)$ converges to a point $(\mathbf{W}^*, \mathbf{b}^*, \mathbf{z}^*, \mathbf{a}^*)$. Because $F$ is continuous, $\lim_{k \to \infty}(\mathbf{W}^k, \mathbf{b}^k, \mathbf{z}^k, \mathbf{a}^k) = F(\lim_{k \to \infty} \mathbf{W}^k, \lim_{k \to \infty} \mathbf{b}^k, \lim_{k \to \infty} \mathbf{z}^k, \lim_{k \to \infty} \mathbf{a}^k) = F(\mathbf{W}^*, \mathbf{b}^*, \mathbf{z}^*, \mathbf{a}^*)$.

(b). Because $(\mathbf{W}^k, \mathbf{b}^k, \mathbf{z}^k, \mathbf{a}^k)$ is convergent and a convergent sequence is bounded, therefore it is also bounded. $\qquad \square$

## E  PROOF OF LEMMA 3

*Proof.* As shown in (Wang et al. (2015); Xu & Yin (2013)),

$$\partial_{\mathbf{W}^{k+1}} F = \{\partial_{W_1^{k+1}} F\} \times \{\partial_{W_2^{k+1}} F\} \times \cdots \times \{\partial_{W_L^{k+1}} F\}.$$

$$\nabla_{\mathbf{b}^{k+1}} F = \nabla_{b_1^{k+1}} F \times \nabla_{b_2^{k+1}} F \times \cdots \times \nabla_{b_L^{k+1}} F.$$

where $\times$ denotes Cartesian Product.
For $W_l^{k+1}$,

$$\partial_{W_l^{k+1}} F$$

$$= \partial\Omega_l(W_l^{k+1}) + \nabla_{W_l^{k+1}} \phi(a_{l-1}^{k+1}, W_l^{k+1}, b_l^{k+1}, z_l^{k+1})$$

(Definition of $F$ in Problem 2)

$$= \nabla_{W_l^{k+1}} \phi(a_{l-1}^{k+1}, W_l^{k+1}, b_l^{k+1}, z_l^{k+1})$$
$$- \nabla_{W_l^k} \phi(a_{l-1}^{k+1}, W_l^k, b_l^k, z_l^k) - \theta_l^{k+1} \circ (W_l^{k+1} - W_l^k)$$
$$+ \partial\Omega_l(W_l^{k+1}) + \nabla_{W_l^k} \phi(a_{l-1}^{k+1}, W_l^k, b_l^k, z_l^k)$$
$$+ \theta_l^{k+1} \circ (W_l^{k+1} - W_l^k)$$
$$= \rho(W_l^{k+1} - W_l^k) a_{l-1}^{k+1} (a_{l-1}^{k+1})^T + \rho(b_l^{k+1} - b_l^k)(a_{l-1}^{k+1})^T$$
$$- \rho(z_l^{k+1} - z_l^k)(a_{l-1}^{k+1})^T - \theta_l^{k+1} \circ (W_l^{k+1} - W_l^k)$$
$$+ \partial\Omega_l(W_l^{k+1}) + \nabla_{W_l^k} \phi(a_{l-1}^{k+1}, W_l^k, b_l^k, z_l^k)$$
$$+ \theta_l^{k+1} \circ (W_l^{k+1} - W_l^k)$$

On one hand, we have

$$\|\rho(W_l^{k+1} - W_l^k) a_{l-1}^{k+1} (a_{l-1}^{k+1})^T + \rho(b_l^{k+1} - b_l^k)(a_{l-1}^{k+1})^T$$
$$- \rho(z_l^{k+1} - z_l^k)(a_{l-1}^{k+1})^T - \theta_l^{k+1} \circ (W_l^{k+1} - W_l^k)\|$$
$$\leq \rho\|(W_l^{k+1} - W_l^k) a_{l-1}^{k+1} (a_{l-1}^{k+1})^T\| + \rho\|(b_l^{k+1} - b_l^k)(a_{l-1}^{k+1})^T\|$$
$$+ \rho\|(z_l^{k+1} - z_l^k)(a_{l-1}^{k+1})^T\| + \|\theta_l^{k+1} \circ (W_l^{k+1} - W_l^k)\| \text{(triangle inequality)}$$
$$\leq \rho\|W_l^{k+1} - W_l^k\|\|a_{l-1}^{k+1}\|\|a_{l-1}^{k+1}\| + \rho\|b_l^{k+1} - b_l^k\|\|a_{l-1}^{k+1}\|$$
$$+ \rho\|z_l^{k+1} - z_l^k\|\|a_{l-1}^{k+1}\| + \|\theta_l^{k+1}\|\|W_l^{k+1} - W_l^k\|$$

(Cauchy-Schwarz inequality)

$$\leq \rho M_{\mathbf{a}}\|b_l^{k+1} - b_l^k\| + \rho M_{\mathbf{a}}\|z_l^{k+1} - z_l^k\|$$
$$(\rho M_{\mathbf{a}}^2 + \|\theta_l^{k+1}\|)\|W_l^{k+1} - W_l^k\| \text{(Lemma 2)}$$

On the other hand, the optimality condition of equation 2 yields

$$0 \in \partial\Omega_l(W_l^{k+1}) + \nabla_{W_l^k} \phi(a_{l-1}^{k+1}, W_l^k, b_l^k, z_l^k) + \theta_l^{k+1} \circ (W_l^{k+1} - W_l^k)$$

Therefore, there exists $g_{1,l}^{k+1} \in \partial_{W_l^{k+1}} F$ such that

$$\|g_{1,l}^{k+1}\| \leq \rho M_{\mathbf{a}}\|b_l^{k+1} - b_l^k\| + \rho M_{\mathbf{a}}\|z_l^{k+1} - z_l^k\|$$
$$+ (\rho M_{\mathbf{a}}^2 + \|\theta_l^{k+1}\|)\|W_l^{k+1} - W_l^k\|$$

This shows that there exists $g_1^{k+1} = g_{1,1}^{k+1} \times g_{1,2}^{k+1} \times \cdots \times g_{1,L}^{k+1} \in \partial_{\mathbf{W}^{k+1}} F$ and $C = \max(\rho M_{\mathbf{a}}, \rho M_{\mathbf{a}}^2 + \|\theta_1^{k+1}\|, \rho M_{\mathbf{a}}^2 + \|\theta_2^{k+1}\|, \cdots, \rho M_{\mathbf{a}}^2 + \|\theta_L^{k+1}\|)$ such that

$$\|g_l^{k+1}\| \leq C(\|\mathbf{W}^{k+1} - \mathbf{W}^k\| + \|\mathbf{b}^{k+1} - \mathbf{b}^k\|$$
$$+ \|\mathbf{z}^{k+1} - \mathbf{z}^k\|)$$

Similarly, for $b_l^{k+1}$,

$$
\begin{aligned}
\nabla F_{b_l^{k+1}} &= \nabla_{b_l^{k+1}}\phi(a_{l-1}^{k+1}, W_{l-1}^{k+1}, b_l^{k+1}, z_l^{k+1}) \\
&= \nabla_{b_l^{k+1}}\phi(a_{l-1}^{k+1}, W_{l-1}^{k+1}, b_l^{k+1}, z_l^{k+1}) \\
&\quad - \nabla_{b_l^k}\phi(a_{l-1}^{k+1}, W_{l-1}^{k+1}, b_l^k, z_l^k) - \rho(b_l^{k+1} - b_l^k) \\
&\quad (\nabla_{b_l^k}\phi(a_{l-1}^{k+1}, W_{l-1}^{k+1}, b_l^k, z_l^k) + \rho(b_l^{k+1} - b_l^k)) = 0 \text{ by} \\
&\quad \text{the optimality condition of equation 3)} \\
&= \rho(z_l^{k+1} - z_l^k).
\end{aligned}
$$

Therefore, there exists $g_{2,l}^{k+1} = \nabla_{b_l^{k+1}} F$ such that

$$
\|g_{2,l}^{k+1}\| = \rho\|z_l^{k+1} - z_l^k\|
$$

This shows that there exists $g_2^{k+1} = g_{2,1}^{k+1} \times g_{2,2}^{k+1} \times \cdots \times g_{2,L}^{k+1} = \nabla_{\mathbf{b}^{k+1}} F$ such that

$$
\|g_2^{k+1}\| = \rho\|\mathbf{z}^{k+1} - \mathbf{z}^k\|.
$$

$\square$

.

## F  PROOF OF THEOREM 2

*Proof.* To prove this theorem, we will first show that $c_k$ satisfies two conditions: (1). $c_k \geq c_{k+1}$. (2). $\sum_{k=0}^{\infty} c_k$ is bounded. We then conclude the convergence rate of $o(1/k)$ based on these two conditions. Specifically, first, we have

$$
\begin{aligned}
c_k &= \min_{0 \leq i \leq k}\Big(\sum_{l=1}^{L} \|\theta_l^{i+1} \circ (W_l^{i+1} - W_l^i)^{\circ 2}\|_1/2 \\
&\quad + (\rho/2)\sum_{l=1}^{L} \|b_l^{i+1} - b_l^i\|_2^2 + (\rho/2)\sum_{l=1}^{L} \|z_l^{i+1} - z_l^i\|_2^2 \\
&\quad + \sum_{l=1}^{L-1} \|\tau_l^{i+1} \circ (a_l^{i+1} - a_l^i)^{\circ 2}\|_1/2\Big) \\
&\geq \min_{0 \leq i \leq k+1}\Big(\sum_{l=1}^{L} \|\theta_l^{i+1} \circ (W_l^{i+1} - W_l^i)^{\circ 2}\|_1/2 \\
&\quad + (\rho/2)\sum_{l=1}^{L} \|b_l^{i+1} - b_l^i\|_2^2 + (\rho/2)\sum_{l=1}^{L} \|z_l^{i+1} - z_l^i\|_2^2 \\
&\quad + \sum_{l=1}^{L-1} \|\tau_l^{i+1} \circ (a_l^{i+1} - a_l^i)^{\circ 2}\|_1/2\Big) \\
&= c_{k+1}
\end{aligned}
$$

Therefore $c_k$ satisfies the first condition. Second,

$$
\begin{aligned}
&\sum_{k=0}^{\infty} c_k \\
&= \sum_{k=0}^{\infty} \min_{0 \leq i \leq k}\Big(\sum_{l=1}^{L} \|\theta_l^{i+1} \circ (W_l^{i+1} - W_l^i)^{\circ 2}\|_1/2 \\
&\quad + (\rho/2)\sum_{l=1}^{L} \|b_l^{i+1} - b_l^i\|_2^2 + (\rho/2)\sum_{l=1}^{L} \|z_l^{i+1} - z_l^i\|_2^2 \\
&\quad + \sum_{l=1}^{L-1} \|\tau_l^{i+1} \circ (a_l^{i+1} - a_l^i)^{\circ 2}\|_1/2\Big) \\
&\leq \sum_{k=0}^{\infty}\Big(\sum_{l=1}^{L} \|\theta_l^{k+1} \circ (W_l^{k+1} - W_l^k)^{\circ 2}\|_1/2 \\
&\quad + (\rho/2)\sum_{l=1}^{L} \|b_l^{k+1} - b_l^k\|_2^2 + (\rho/2)\sum_{l=1}^{L} \|z_l^{k+1} - z_l^k\|_2^2 \\
&\quad + \sum_{l=1}^{L-1} \|\tau_l^{k+1} \circ (a_l^{k+1} - a_l^k)^{\circ 2}\|_1/2\Big) \\
&\leq F(\mathbf{W}^0, \mathbf{b}^0, \mathbf{z}^0, \mathbf{a}^0) - F(\mathbf{W}^*, \mathbf{b}^*, \mathbf{z}^*, \mathbf{a}^*) \text{(Lemma 1)}
\end{aligned}
$$

So $\sum_{k=0}^{\infty} c_k$ is bounded and $c_k$ satisfies the second condition. Finally, it has been proved that the sufficient conditions of convergence rate $o(1/k)$ are: (1) $c_k \geq c_{k+1}$, and (2) $\sum_{k=0}^{\infty} c_k$ is bounded, and (3) $c_k \geq 0$ (Lemma 1.2 in (Deng et al. (2017))). Since we have proved the first two conditions and the third one $c_k \geq 0$ is obvious, the convergence rate of $o(1/k)$ is proven. $\square$

## G    DISCUSSION

We discuss convergence conditions of our DLAM compared with SGD-type methods and the ADMM method. The comparison demonstrates that our convergence conditions are more general than others.

### G.1    DLAM VERSUS SGD

One influential work by Ghadimi et al. (Ghadimi & Lan (2016)) guaranteed that the SGD converges to a critical point, which is similar to our convergence results. While the SGD requires the objective function to be Lipschitz differentiable, bounded from below (Ghadimi & Lan (2016)), our DLAM allows for non-smooth functions such as Relu. Therefore, our convergence conditions are milder than SGD.

### G.2    DLAM VERSUS ADMM

For ADMM, Zeng et al. showed that the ADMM is convergent to a critical point with a sublinear convergence rate $O(1/k)$ (Zeng et al. (2019)), which is similar to our convergence results. However, the ADMM requires activation functions $h_l(\bullet)$ to be twice-differentiable bounded, whereas our DLAM allows $h_l(\bullet)$ to be non-smooth. This also shows that the assumptions of our DLAM are more general than those of the ADMM.

## H    RELATED WORK

All of the existing works on optimization methods in deep neural network problems falls into two major classes: stochastic gradient descent methods and alternating minimization methods. This research related to both is discussed in this section.

**Stochastic gradient descent methods:** The renaissance of SGD can be traced back to 1951 when Robbins and Monro published the seminal paper (Robbins & Monro (1951)). The famous back-propagation algorithm was introduced by Rumelhart et al. (Rumelhart et al. (1986)). Many variants of SGD methods have since been presented, including the use of Polyak momentum, which accelerates the convergence of iterative methods (Polyak (1964)), and research by Sutskever et al., who highlighted the importance of Nesterov momentum and initialization (Sutskever et al. (2013)). Many well-known SGD methods that incorporate with adaptive learning rates have been proposed by the deep learning community, including AdaGrad (Duchi et al. (2011)), RMSProp (Tieleman & Hinton), Adam (Kingma & Ba (2014)) and AMSGrad (Reddi et al. (2018)).

**Alternating minimization methods for deep learning:** Previous work on the application of alternating minimization algorithms to deep learning problems can be categorized into two main types. The first research strand proposes the use of alternating minimization algorithms for specific applications. For example, Taylor et al. and Wang et al. presented an Alternating Direction Method of Multipliers (ADMM) algorithm to transform a fully-connected neural network problem into an equality-constrained problem, where many subproblems split by ADMM can be solved in parallel (Taylor et al. (2016); Wang et al. (2019)), while Zhang et al. handled very deep supervised hashing (VDSH) problems by utilizing an ADMM algorithm to overcome issues related to vanishing gradients and poor computational efficiency (Zhang et al. (2016)). Zhang and Bastiaan trained a deep neural network by utilizing ADMM with a graph (Zhang & Kleijn (2017)) and Askari et al. introduced a new framework for multilayer feedforward neural networks and solved the new framework using block coordinate descent (BCD) methods (Askari et al. (2018)). Others have proposed novel alternating minimization methods and proved their convergence results. For instance, Carreira and Wang suggested a method involving the use of auxiliary coordinates (MAC) to replace a nested neural network with a constrained problem without nesting (Carreira-Perpinan & Wang (2014)). Zeng et al. and Lau et al. both proposed BCD algorithms, proving its convergence via the Kurdyka-ojasiewicz (KL) property (Jinshan Zeng (2018); Lau et al. (2018)), while Choromanska et al. proposed a BCD algorithm for training deep feedforward neural networks based on the concept of co-activation memory (Choromanska et al. (2018)) and a BCD algorithm with R-linear convergence was proposed by Zhang and Brand to train Tikhonov regularized deep neural networks (Zhang & Brand (2017)). However, most of these researchers focused on specific applications of neural networks rather than their general formulations. Even though several do discuss general neural network problems and provide theoretical guarantees, the assumptions involved are hard to satisfy in practice.

