# OpenReview forum: "Gradient-free Neural Network Training by Multi-convex Alternating Optimization"
_ICLR.cc/2020/Conference — Reject_

### Official Review · AnonReviewer3 · 2019-10-22
**Official Blind Review #3**

**Rating:** 1

**Review:**

This paper proposes Deep Learning Alternating Minimization (DLAM) algorithm. First, deep learning optimization problems are formulated as multi-convex Problem 2, by introducing additional variables, constraints and relaxations. Second, alternating update is then used to solve Problem 2, the analysis shows that weights of update converge to a critical point with O(1/k) rate. Finally, some experiments are conducted on MNIST and Fashion-MNIST to show that DLAM is better than SGD, Adagrad, Adadelta and ADMM.

1. DLAM follows dlADMM in Wang et al. (2019), and the only difference between Problem 2 and Problem 2 in Wang et al. (2019) is the indicator function rather than squared l2 loss for a_l. The update is quite similar with dlADMM, with the same convergence result. The technical contribution is incremental.

2. The author claim that Problem 2 is multi-convex. However, I did not see why this is a good point. First, multi-convexity does not imply that gradient update can find global optimum. Second, the results presented here is standard, i.e., O(1/k) convergence to a critical point, which does not show any advantage over Problem 2 in Wang et al. (2019).

3. The experiments are also questionable.
a) The most related dlADMM (Wang et al. (2019)) is not compared here, which can not empirically show why Problem 2 here is a better choice than dlADMM.
b) The accuracy on MNIST is pretty low. In Wang et al. (2019), the results are about 0.9x, here only 0.7x. Why? And such low accuracy on MNIST seems not convincing to claim "convergence".
c) Please compare with Adam.
d) What is the ADMM update in the experiments? Directly use ADMM on Problem 1 or 2?

4. The term "global convergence" seems misleading. It is not convergence to global optima.

Overall, I found the formulation quite similar with dlADMM (Wang et al. (2019)), the contribution is incremental, the analysis did not show why multi-convex formulation Problem 2 is good (the rate is the same as standard results), and the experiments are questionable and also did not show advantages of the formulation and proposed method.


=======Update=======
Thanks for the rebuttal. I keep my rating since there is no updated version of the paper to address my concerns.


**Experience Assessment:**

I have read many papers in this area.

**Review Assessment: Checking Correctness Of Derivations And Theory:**

I assessed the sensibility of the derivations and theory.

**Review Assessment: Checking Correctness Of Experiments:**

I carefully checked the experiments.

**Review Assessment: Thoroughness In Paper Reading:**

I read the paper thoroughly.

---

> ### Author Response · Authors · 2019-11-15
> **Thank You for the Feedback**
>
> Dear Reviewer:
>                Thank you so much for your suggestions. Our answers are shown as follows:
> Q1. I don’t understand why Problem 2 is multi-convex is a good point.
> A1. The multi-convexity of Problem 2 makes every subproblem convex and solvable. Otherwise, for the sigmoid and tanh activation functions, the z-subproblem can not be solved exactly in the dlADMM algorithm, and a lookup table is required.
> Q2. The experiments are also questionable.
> a) The most related dlADMM (Wang et al. (2019)) is not compared here, which can not empirically show why Problem 2 here is a better choice than dlADMM.
> b) The accuracy on MNIST is pretty low. In Wang et al. (2019), the results are about 0.9x, here only 0.7x. Why? And such low accuracy on MNIST seems not convincing to claim "convergence".
> c) Please compare it with Adam.
> d) What is the ADMM update in the experiments? Directly use ADMM on Problem 1 or 2?
> A2.
> a) We may compare our DLAM with dlADMM in the future version.
> b). In Figure 2(c), we show that the accuracy of the DLAM is above 0.8 for the MNIST dataset, and the actual accuracy is  0.91.
> c). We may compare our DLAM with Adam in the future version.
> d). The ADMM in the experiments is detailed in the reference Taylor et al. (2016) in our original paper. It applies ADMM neither on Problem 1 or Problem 2, instead, the ADMM is applied to the relaxed problem of Problem 1 in the reference.
> Q3. The term "global convergence" seems misleading. It is not convergence to global optima.
>
> A3. The term “global convergence” refers to the convergence however parameters are initialized, it does not mean the convergence to the global optima. [1]
> Reference:
> 1. https://en.wikipedia.org/wiki/Local_convergence.

---

### Official Review · AnonReviewer1 · 2019-10-30
**Official Blind Review #1**

**Rating:** 6

**Review:**

The motivation for this paper is as follows:

Why SGD:
1. Easy
2. applied in online settings

However:
1. Convergence proof has been done by others, but the assumptions of their proofs cannot be applied to problems involving deep neural networks, which are highly nonsmooth and nonconvex
2. Suffer from gradient vanish
3. sensitive to the input

The paper is based on new ideas  on alternative minimization method. In particular the  loss function is reformulated as nested function associated with multiple linear and nonlinear transformations. This nested structure is then decomposed into a series of linear and nonlinear equality constraints by introducing auxiliary variables and penalty hyperparameters. Then  multiple subproblems are generated which can be minimized alternatively. (using ADMM, BCD)
However, these methods suffer from problems:
1. Convergence properties are sensitive to penalty parameters.
2. Lack of unified theoretical frameworks with general conditions. (Not fully proved the
convergence. Most of the work is based on some assumptions)


Main Assumption: Activation functions are quasilinear functions.


pros (+):
1. Generic, easy to extend it to convolutional layers and recurrent layers.

2. Transform the Neural network optimization problem into inequality constrained
problem (new point of view)

3. Ensure convexity of subproblems. (quadractic approximation of activation function).
4. Ensure global minima, and converges to the critical point (where 0 is in the set of
partial differential of F(F(W,b,,z,a) is the augment loss function) with respect to W*,
and set of partial differential of F with respect to b*).
5. The convergence rate is O(1/k), k is the number of iterations of updating (W,b,z,a).
6. inequality-constraint prevents the output of a nonlinear function from changing much
and reduces sensitivity to the input.
7. Matrix inversion is avoided by quadratic approximation
8. No need strict and complex condition, such as KL properties to prove convergence.
Instead, this algorithm needs simple and mild conditions to guarantee convergence,
and cover most of the loss function and activation functions.
9. The choice of hyperparameters has no effect on convergence.
10. Much better performance on MNIST

cons(-):
1. penalty parameters need to be tuned as ADMM does.

2. The regularizations $\omega$ is l1 and l2, otherwise, it will not be closed-form
solutions. The regularization like dropout, batch-norm can not be added into this algorithm.

3. The dataset is relatively naive. Only MNIST and Fashion MNIST. It would be better to include little more sophisticate dataset like ImageNet or Cifar10. Since the computation complexity is O(d^2) where d is the dimension of the features, large scale of image like ImageNet could cause problem. (Comparison: features of MNIST: 196, features of Fashion MNIST:784, features of ImageNet: 65536).

4. The Adagrad has better performance than DLAM on 500*500 on Fashion MNIST and close performance on MNIST. As the authors suggest: DLAM performed competitively for the Fashion MNIST dataset. This can be interpreted as follows:  the proposed algorithm performs well on easy datasets, however the performance is not much better than other algorithms  on Fashion MNIST, which is little more challenging. What would be your comment about this criticism? Can you provide any theoretical insight?  Certainly  it would strengthen the paper if there were more comparisons of  DLAM on more complex dataset.

**Experience Assessment:**

I have read many papers in this area.

**Review Assessment: Checking Correctness Of Derivations And Theory:**

I did not assess the derivations or theory.

**Review Assessment: Checking Correctness Of Experiments:**

I assessed the sensibility of the experiments.

**Review Assessment: Thoroughness In Paper Reading:**

I read the paper at least twice and used my best judgement in assessing the paper.

---

> ### Author Response · Authors · 2019-11-15
> **Thank you for the Feedback**
>
> Dear Reviewer:
>        Thank you so much for your suggestions. Our answers are shown as follows:
>
> Q1: penalty parameters need to be tuned as ADMM does.
> A1: Yes. The choice of penalty parameters are given in Section 4.2.1.
>
> Q2: The regularization like dropout, batch-norm can not be added into this algorithm.
> A2:  We may discuss how to add dropout and batch-norm in our future version.
>
> Q3: large scale of image like ImageNet could cause problem, Certainly  it would strengthen the paper if there were more comparisons of  DLAM on more complex dataset.
> A3:  We may add the imageNet and other datasets to our paper in the future version.

---

### Decision · Program_Chairs · 2019-12-19

**Decision:**

Reject

**Comment:**

The paper proposes a new learning algorithm for deep neural networks that first reformulates the problem as a multi-convex and then uses an alternating update to solve. The reviewers are concerned about the closeness to previous work, comparisons with related work like dlADMM, and the difficulty of the dataset. While the authors proposed the possibility of addressing some of these issues, the reviewers feel that without actually addressing them, the paper is not yet ready for publication.